# Impact of functional status and biomarkers on hospital costs and readmission rates in geriatric patients: An observational study with comprehensive geriatric assessment

Irene Simonetti [1¤a*], Stefano Landi[2], Chiara Leardini[2], Anna Giani[3], Arianna Bortolani[3], Francesco Fantin[3¤b]

**1** Economics and Business, University of Amsterdam, The Netherlands, **2** Department of Management, University of Verona, Verona, Italy, **3** Section of Geriatric Medicine, Department of Medicine, University of Verona, Verona, Italy,

¤a Research Centre for Longevity Risk (RCLR), University of Amsterdam and Vrije Universiteit Amsterdam, The Netherlands
¤b Section of Geriatric Medicine, Centre for Medical Sciences- CISMed, Department of Psychology and Cognitive Science, University of Trento, Rovereto (TN), Italy
* i.simonetti@uva.nl

## Abstract

This study aimed to analyze the association between functional status, biomarkers, and hospitalization characteristics on costs and the probability of re-admission at 30 and 180 days in geriatric patients. It is used an observational design with both administrative data and additional clinical data not usually collected. Multivariate linear regression for hospitalization costs and multivariate logistic regressions for readmissions were used. Variables studied included the Barthel Index, Charlson index, albumin and blood pressure levels, previous hospitalizations, length of stay (LoS), and controls. Data from 953 patients aged over 65, admitted to the Geriatric ward between September 1st, 2018, and December 31st, 2019, were analyzed. The Charlson comorbidity index, number of comorbidities, and LoS were positively related to hospitalization costs. Previous hospitalizations and LoS were the main predictors of readmission. Systolic blood pressure was negatively associated with the odds of re-admission but showed no association with hospital costs. Higher functional status, as measured by the Barthel index, was linked to lower odds of unplanned hospitalization but was not statistically significant for costs. Functional status and biomarkers had moderate effects on costs and readmission odds. These findings can aid in early healthcare planning and resource management, providing valuable information for prioritizing patients and designing cost-effective care interventions.

## Introduction

The aging of the population in both the Western world and developing countries has a significant impact on healthcare systems. According to the United

**Data availability statement:** Please note that this study involves human research participants, and the data collected includes sensitive health information from elderly patients hospitalized in the Geriatric Unit of the University Hospital of Verona. Due to ethical restrictions and privacy regulations related to the protection of patient confidentiality, imposed by the Ethics Committee for Clinical Trials of the Provinces of Verona and Rovigo, the full dataset cannot be made publicly available. Specifically, the informed consent obtained from participants does not permit public data sharing, and public access to the data would violate institutional and legal data protection policies. However, deidentified data can be made available to qualified researchers upon reasonable request and approval from the aforementioned Ethics Committee. Requests should be directed to the following institutional contact: Comitato Etico per la Sperimentazione Clinica Ufficio di Segreteria Tecnico-Scientifica del Comitato Etico e/o Servizio di Farmacia Borgo Trento – P.le A. Stefani, 1 – 37126 Verona Tel: 0458123236 Email: comitatoetico.veronarovigo@aovr.veneto.it Requests should be directed to the Ethics Committee referring to the study protocol number: "1892CESC".

**Funding:** The author(s) received no specific funding for this work.

**Competing interests:** The authors have declared that no competing interests exist.

Nations report [1], the global population aged 65 and above is projected to increase from 10% in 2022 to 16% by 2050. In Italy, Eurostat projects that by 2050, individuals aged 55 and older will account for 45.9% of the population [2].

The increase in life expectancy and prevalence of age-related pathologies has led to a growing proportion of people getting affected by chronic illnesses and an increase in elderly patients admitted to hospitals. Hospitalization is a serious event for many of these patients due to their increased vulnerability [3]. Geriatric patients often have multiple comorbidities, compromised functional status, and nutritional issues induced by their fragility. Consequently, they are at higher risk for adverse events related to hospitalization [4–6].

Economic studies have shown hospitalization to be one of the main drivers of expenditure in the last years of life [7] and one of the most expensive healthcare services. Patient clinical characteristics can affect the cost for hospitalization [8]. Moreover, frailty and multi-morbidity in geriatric patients lead to multiple hospitalizations and higher readmission rates. Previous studies have highlighted the utility of biomarkers in predicting hospitalization trends and associated healthcare costs in chronic and high-risk populations. For instance [9], demonstrated that patients with elevated NT-proBNP levels had significantly longer hospital stays and higher associated costs.

Healthcare systems worldwide are paying programmatic attention to reducing the rate of hospital readmissions. They are considered an indicator of the quality of care, but they are also an economic burden to healthcare systems, with higher readmissions increasing costs [10].

The demographic transition and population ageing in developed countries are increasing the pressure on healthcare system expenditure. Therefore, the call for effective and value-for-money interventions is of great relevance in geriatric healthcare. To reach this goal, designing health interventions for specific segments of patients that can yield the highest benefit is crucial. Therefore, understanding the predictors of costs and readmissions is critical for targeting elderly patients to develop cost-effective care management interventions.

## The study

To make cost-effective decisions daily, hospital managers and professionals require evidence on how, for example, an increase in comorbidities, loss of functional status, and higher frailty may affect hospital costs and readmissions in elderly patients. Several studies have employed administrative data; however, only a few have included both patient-reported scale and administrative data.

This article contributes to the literature by examining the relationship between patients' clinical characteristics with costs and readmissions in geriatric patients, with particular attention to functional status and biomarkers, by employing a unique dataset that matches collected administrative data (hospital discharge records) with patient variables that are not routinely collected. The current study aims at assessing the effect of functional status and biomarkers such as the Barthel index (BI), the

Charlson index, albumin and blood pressure levels, previous hospitalization and length of stay (LoS) on hospitalization costs and hospital readmissions within 30 days and 6 months.

## Design

An observational retrospective-prospective study was conducted on patients over the age of 65 years who were admitted to the University Hospital of Verona (Italy) between September 2018 and December 2019. A consecutive sampling method was applied in geriatric wards. Patients were eligible to participate in the study if they were aged more than 65 years; were scheduled for an expected hospital stay longer than 24 h; and were conscious, cooperative, and capable of providing written informed consent.

All patients aged more than 65 years who were consecutively admitted to the Geriatric Ward from September 1st, 2018, to December 31st, 2019, were included in the study. In total, 953 patients were admitted. Among them, 169 were excluded from the analysis, since 12 were readmitted less than 48 h after discharge, 59 died during index hospitalization, 97 were transferred to the long-term care unit annexed to the ward, and 1 was transferred to hospice. The total study population, therefore, comprised 784 patients, among which 389 were male and 395 were female. Due to the inadequately short length of the follow-up period, we further excluded 7 patients who died within 5 days after discharge and 11 more due to missing information on hospitalization costs or mis-codification (i.e., patients codified with MDC = pregnancy). The final sample population was composed of 766 patients (Fig 1). Upon admission, all the patients received a complete clinical evaluation that encompassed their detailed clinical history, including pathological conditions, previous admissions, drug regimen, and the occurrence of any adverse drug reaction. Medical care was offered to the participants according to the national and international standards of clinical practice related to the specific medical condition.

## Data

The demographical and clinical data, medical diagnoses, and data on hospital admission were retrieved from hospital discharge records (Schede di Dimissione Ospedaliera, SDO). In Italy, the SDO is an ordinary administrative tool for collecting information on hospitalization (sociodemographic, admission, and clinical characteristics), utilized in both public and private facilities across the nation. Moreover, a dataset with additional clinical scales and patient-reported measures was matched with the administrative data. The study was approved by the Ethics Committee of the University of Verona Hospital, and it was conducted in accordance with the latest revision of the Helsinki Declaration.

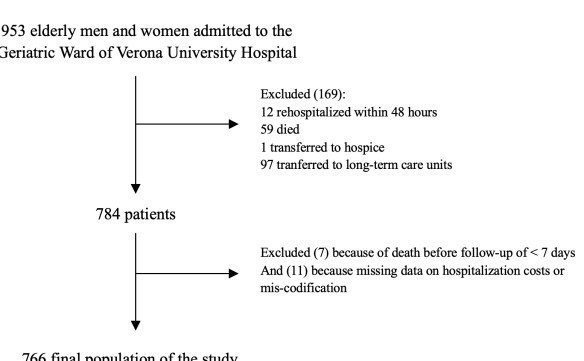

**Fig 1. Flowchart of the study.**

## Outcomes

The primary outcome of our study was to predict hospitalization costs (Euro). For each patient, expenses were ascribed according to the discharge diagnosis-related group (DRG) codes—a system utilized to calculate the amount reimbursed to the hospital by the healthcare purchaser (in this case, the Veneto region). The secondary set of outcomes concerns the patients' short- and long-term unplanned readmission at 30 days and 6 months after hospital discharge.

## Predictors of hospital costs and health outcomes

Many factors can be considered potential determinants (predictors) of hospitalization costs and future readmissions. To enrich the analysis, besides common demographic characteristics such as age and gender, we included clinical variables such as LoS, a dichotomous indicator signaling previous hospitalizations in the past 6 months (following Franchi et al. [11]), the number of diagnoses, dummy variables for the main diagnosis, the Charlson Index, and the BI [12].

We identified the main diagnoses according to the Major Diagnostic Categories (MDC) associated with the DRG system and classified them into seven categories: 1) Circulatory System Disorders (MDC = 2); 2) Digestive System Disorders (MDC = 3); 3) Hepatobiliary and Kidney Disorders (MDC = 4/5); 4) Respiratory Disorders (MDC = 8); 5) Nervous and Mental Disorders (MDC = 11/15); 6) Infectious Disorders (MDC = 13); and 7) Other Disorders (MDC = 1/6/7/9/14). Based on their frequency in the data and the average hospitalization costs, we consider "Respiratory Disorders" as our reference category. Comorbidities were evaluated by using the Charlson comorbidity index (CCI) [13]. The CCI is a weighted index validated to predict 1-year mortality for patients with specific comorbid conditions. It considers the number as well as the seriousness of each comorbid disease, thus providing a composite score that includes the number of diagnoses and the severity score. The Charlson comorbidity score is commonly utilized to assess the prognostic burden of comorbid diseases. This study used the Deyo adaptation coding algorithm for ICD-9-CM data [14,15]. A comorbidity index was also computed by counting the number of patients' pathologies.

Performance in activities of daily life, with special attention to the patients' autonomy in self-care and their mobility level, was assessed through the BI [12]. The index enquirers after 10 primary daily self-care activities (feeding, bathing, grooming, dressing, bowel control, bladder control, toilet use, transfer from bed to chair and back, mobility on level surfaces, and climbing stairs) and identifies each with three levels of autonomy (none, partial, and complete; scored 0, 5, and 10/15, respectively). The sum of the individual scores ranges from 0 (complete dependency) to 100 (complete autonomy). In the present study, we followed the categorizationmethod proposed by Ocagli et al. [16], where we distinguish between "severe dependence" (0–50), "moderate dependence" (51–75), and "mild dependence/independence" (76–100).

Only a few studies have considered the role of specific blood parameters such as serum albumin and blood pressure levels as predictors of hospitalization costs and future (unplanned) readmissions. In the present study, blood pressure and serum albumin were collected at admission and discharge. They were categorized according to common medical thresholds. Based on the entry level of serum albumin, we grouped patients with a "hypoalbuminemia" condition if their serum concentration was less than or equal to 30 g/l [17,18].

Hypoalbuminemia can suggest poor nutritional status or protein malabsorption [19]. Several studies focused on different clinical settings and major diseases have demonstrated a significant relationship between a low level of albumin and increased morbidity [20,21]. Finally, we considered the systolic blood pressure at admission and discharge and grouped patients having values below or above the threshold of 115 mmHg [22].

## Ethical considerations

All participants gave their written, informed consent to participate. The study was carried out following the tenets of the Helsinki Declaration and approved by the local Ethics Committee of the University of Verona Hospital (Prog. 1892CESC). The data has been accessible since November 1st, 2021.

## Results

### Descriptive

Table 1 presents the main characteristics of the study population, divided by sex. Table 2 shows baseline descriptive statistics of our main predictors (column a), besides the univariate (unadjusted) analysis with respect to the logarithm of hospitalization costs (column b). After the additional deletion steps mentioned in the design section, 766 geriatric patients were included in the analysis. The patients' ages ranged between 62 and 107 years, with a median of 85 years (IQR 80–90), while approximately half of the population constituted women (50.5%).

Regarding the clinical predictors, the median LoS in the hospital was 9 days (IQR 7–13), approximately 76% of the patients presented at least one or more comorbidities at admission, and the discharge diagnoses were mostly related to circulatory or respiratory system diseases (32.3% and 26.5%, respectively). According to the previously explained categorization using the BI, half of the geriatric patients were found to have a severe dependence, that is, they received a score of (both at admission and discharge) greater than 50. Finally, for specific clinical parameters, namely the albumin and systolic pressure at admission, the median values were 32 g/l and 130 mmHg, respectively. The univariate analysis with respect to hospitalization costs (column b) revealed that clinical variables such as the LoS, Charlson index, and number of diseases were positively associated with increasing hospitalization costs. As expected, the BI at discharge appeared inversely correlated with costs: the lower the BI score, meaning higher dependence, the higher the costs. Clinical predictors—previous hospitalizations within six months, in particular—were found to positively affect readmission (*unadjusted* OR > 1). The results concerning blood parameters demonstrated a positive relationship between costs and rehospitalizations among people with a low level of albumin (compared with the baseline, > 30 g/l) and low systolic pressure (evaluated either at admission or discharge).

### Multivariable analysis

Table 3 presents the results of the adjusted multivariable regression models. First, we investigated inpatient hospitalization costs (column *a)*. The hospitalization costs ranged between 841€ and 31,066€, which were the minimum and the

**Table 1. Study population.**

|  | Total [Average+sd] (n=784) | Men [Average+sd] (n=389) | Women [Average+sd] (n=395) | p Value |
|---|---|---|---|---|
| **Age (at admission)** | 84,48±6,76 | 83,50±6,82 | 85,45±6,56 | 0,001 |
| **Weight (Kg)** | 68,25±15,35 | 73,12±14,72 | 63,4982±14,45 | 0,001 |
| **Albumin (g/L)** | 31,33±5,50 | 31,32±5,23 | 31,34±5,75 | 0,948 |
| **Systolic Blood Pressure (mmHg)** | 128,32±19,52 | 127,32±19,39 | 129,29±19,62 | 0,157 |
| **Diastolic Blood Pressure (mmHg)** | 72,26±10,30 | 71,68±10,30 | 72,83±10,28 | 0,118 |
| **MAP[a] (mmHg)** | 90,94±12,14 | 90,23±12,03 | 91,65±12,22 | 0,101 |
| **PP[b] (mmHg)** | 56,06±15,04 | 55,64±15,18 | 56,47±14,91 | 0,444 |
| **HR[c] (bpm)** | 81,34±16,12 | 78,82±15,66 | 83,81±16,21 | 0,001 |
| **Length of stay (days)** | 11,32±7,65 | 11,64±8,65 | 10,99±6,52 | 0,235 |
| **BI[d]** | 46,88±32,90 | 52,32±34,07 | 41,47±30,80 | 0,001 |
| **CCI[e]** | 6,02±1,97 | 6,20±2,16 | 5,85±1,75 | 0,013 |

[a]MAP = Mean arterial pressure.

[b]PP = Pulse Pressure.

[c]HR = Heart Rate.

[d]BI = Barthel index.

[e]CCI = Charlson comorbidity index.

**Table 2. Predictors of hospitalization costs (univariate analysis).**

|  | Median/Freq. | *Log*(HospitalCosts) |
|---|---|---|
|  | (a) | (b) |
| **Age** | 85 (80-90) |  |
| **Age groups (≤ 84, baseline):** |  |  |
| **85–89** | 197 (25.7) | −0.062 |
| **90+** | 198 (25.9) | −0.061 |
| **Female** | 387 (50.5) | −0053* |
| **Hospital. pre-6months:** |  |  |
| **Yes** | 141(18.4) | 0.043 |
| **Length of stay** | 9 (7-13) | 0.342*** |
| **Charlson index** | 1 (1-2) |  |
| **Charlson index (=0, baseline):** |  |  |
| **≥1** | 581 (75.9) | 0.225*** |
| **Num. diagnoses** | 4 (3-5) | 0.092*** |
| **Main Diagnosis:** |  |  |
| **Circulatory Disorders** | 247 (32.3) | −0.054 |
| **Digestive Disorders** | 70 (9.1) | −0.016 |
| **Respiratory Disorders** | 203 (26.5) | −0.092* |
| **Hepatob. & Kidney Dis.** | 75 (9.8) | 0.043 |
| **Nervous & Mental Dis.** | 37 (4.8) | −0.166** |
| **Infectious Disorders** | 76 (9.9) | 0.582*** |
| **Other Disorders** | 58 (7.6) | −0.452*** |
| **Barthel index (at admission)** | 45 (15-75) |  |
| **Barthel index (at admission, ≤ 50):** |  |  |
| **51 - 75** | 164 (21.4) | −0.022 |
| **76+** | 175 (22.9) | −0.060 |
| **Barthel index (at discharge)** | 50 (15-85) | −0.002*** |
| **Barthel index (at discharge, ≤ 50):** |  |  |
| **51–75** | 134 (17.5) | −0.088** |
| **76+** | 226 (29.5) | −0.131*** |
| **Albumin** | 32 (28-35) |  |
| **Albumin (baseline: > 30 g/l):** |  |  |
| **≤ 30 g/l** | 299 (38.7) | 0.103*** |
| **Systolic Pressure (at admission)** | 130 (115-140) |  |
| **Systolic Pressure (at admission)** |  |  |
| **≤115 mmHg** | 208 (27.2) | 0.098** |
| **Systolic Pressure (at discharge):** | 120 (110-130) |  |
| **Systolic Pressure (at discharge):** |  |  |
| **≤115 mmHg** | 231 (30.2) | 0.085** |

Data in column (a) are presented as number of observations (plus the corresponding %), or median (plus the interquartile range).

T-test are performed: *** $p < .01$, ** $p < .05$, * $p < .1$

maximum cost per patient observed in the sample, respectively, with a median value equal to 3285€. Interestingly, only clinical variables seemed to significantly determine costs. With all the other variables held constant, the LoS roughly accounted for a 24% increase for a 10% increase in duration (95% CI: 19.3–28.9). Considering the median value of

**Table 3. Results – Predictors of hospital costs and re-admissions.**

| | Log(Hospital Costs) | Re-admission (OR) | |
|---|---|---|---|
| | (a) | (b) | (c) |
| | | 30days | 6 months |
| **Age groups (<84, baseline):** | | | |
| **85–89** | −0.041 | 1.094 | 0.945 |
| | (0.031) | (0.287) | (0.190) |
| **90+** | −0.027 | 0.854 | 0.833 |
| | (0.032) | (0.240) | (0.177) |
| **Female** | −0.025 | 1.220 | 1.103 |
| | (0.026) | (0.271) | (0.184) |
| **Previous hospital. (6 months earlier)** | 0.011 | 4.549*** | 4.778*** |
| | (0.033) | (1.046) | (1.005) |
| **Log(Length of stay)** | 0.241*** | 1.331 | 1.589** |
| | (0.024) | (0.279) | (0.253) |
| **Charlson index** | 0.044*** | 1.002 | 1.097 |
| | (0.009) | (0.074) | (0.061) |
| **Number of diagnoses** | 0.032*** | 1.082 | 0.998 |
| | (0.009) | (0.080) | (0.056) |
| **Respiratory Disorders, (baseline):** | | | |
| **Circulatory System Disorders** | −0.039 | 1.291 | 1.604* |
| | (0.034) | (0.373) | (0.348) |
| **Digestive System Disorders** | −0.041 | 1.384 | 1.538 |
| | (0.048) | (0.557) | (0.474) |
| **Hepatobiliary & Kidney Disorders** | −0.065 | 1.041 | 1.326 |
| | (0.047) | (0.422) | (0.402) |
| **Nervous and Mental Disorders** | −0.170** | 0.657 | 0.841 |
| | (0.062) | (0.431) | (0.365) |
| **Infectious Disorders** | 0.437*** | 1.074 | 1.007 |
| | (0.047) | (0.427) | (0.310) |
| **Other Disorders** | −0.377*** | 1.431 | 1.571 |
| | (0.052) | (0.634) | (0.523) |
| **Barthel index category, (<=50, baseline):** | | | |
| **51 - 75** | 0.011 | 0.585 | 0.796 |
| | (0.033) | (0.186) | (0.188) |
| **76+** | 0.011 | 0.552* | 1.095 |
| | (0.035) | (0.167) | (0.234) |
| **Albumin category, (baseline: > 30g/l):** | | | |
| **<=30 g/l** | −0.006 | 1.039 | 1.196 |
| | (0.027) | (0.239) | (0.207) |
| **Systolic Blood pressure, (baseline: > 115 mmHg):** | | | |
| **<=115 mmHg)** | −0.035 | 1.398 | 1.533* |
| | (0.029) | (0.319) | (0.269) |
| **Observations** | 766 | 766 | 766 |
| **R-squared** | 0.409 | – | – |
| **Pseudo-R2** | – | 0.106 | 0.098 |

Logit estimates for readmission and mortality outcomes; odd ratios are reported.

For the primary outcome "hospital costs", the Barthel index and the systolic blood pressure are considered at admission, while for readmission outcomes at discharge.

*** $p < .01$, ** $p < .05$, * $p < .1$.

hospitalization costs, in monetary terms, this elasticity was approximately 792€. The effects of any additional coexisting medical conditions (i.e., comorbidities)—measured by the number of diagnoses at discharge and the Charlson index— were found to be, although mild, good predictors of costs. They accounted for an increase in costs by approximately 3.1% and 4.4%, respectively, amounting to expenditures between 100€ and 145€. Looking at the type of diagnosis revealed that significantly higher costs were primarily related to patients with infections than to patients with a respiratory disorder $((e^{0.437} - 1) * 100\% = 55\%)$.

In monetary terms, based on the median cost of approximately 3670€ among those who had been diagnosed with a respiratory disease, patients with infections accounted for about 2019€ higher costs. On the contrary, individuals with nervous and mental disorders showed 18.5% lower costs (approximately 679€ less). No statistically significant differences were found regarding cardiovascular diseases, among neither the blood parameters nor the functional status variable.

Columns *b* to *e* show the multivariate logistic regressions for readmission looking at both short- and long-term time windows. The experience of at least one hospitalization over the past 6 months was observed to increase the probability of readmission within 30 days (and 6 months) by about four times (OR: 4.55 - CI: 2.90-7.14). While patients' LoS seemed to be a good predictor of future rehospitalizations within 6 months, this was not found to be true in the short run. Compared with individuals with high systolic blood pressure at discharge, those with values of greater than 115 mmHg showed an increased likelihood of future hospitalizations (OR: 1.533). Compared with patients with a respiratory condition, those with cardiovascular diseases were observed to be more likely to experience future readmissions. Similar results were found when the probability of readmission within a year was considered (S1Table).

## Discussion

This paper investigates the relationship between functional status, biomarkers, and hospitalization characteristics in relation to hospital costs and the likelihood of readmission at 30 days and six months in geriatric patients.

The univariate analysis indicates that hospital costs are positively correlated with both length of stay and the presence of comorbidities (as measured by a Charlson index of ≥1), and the number of pathologies. Among medical diagnoses at discharge, infectious diseases are highly associated with increased hospital costs. The study further confirms that a decline in functional status, reflected by lower Barthel Index (BI) values - especially at discharge – is related to higher costs. Regarding non-elective hospital readmission, a prior hospitalization within six months emerged as the most significant predictor, showing a strong association with readmission at both 30 days and six months post-discharge. Length of stay influences readmission rates, particularly at six months. Lower albumin levels (≤30 g/dl) are linked to an increased likelihood of readmission, while lower systolic blood pressure (≤115 mmHg) is identified as a predictor of readmission at both 30 days and six months. Finally, respiratory disorders are associated with a higher risk of rehospitalization rates at 30 days.

The multivariable regression analyses reveals that the length of stay, the Charlson index, and the number of diagnoses are significant drivers of hospital costs, while a prior hospitalization emerges as a key predictor of a readmission at both 30 days and 6 months. In more detail, the length of stay during the index hospitalization stands out as one of the most critical factors influencing both hospital costs and readmissions within six months. These findings align with previous research [23,24]. While Jencks et al. [24] also reported data on readmissions up to one year, their focus remained on short-term rehospitalization patterns. In contrast, this study expands upon existing literature by incorporating a six-month follow-up, offering a more detailed view of both short- and medium-term readmission risks.

The association between comorbidities and rising healthcare costs is well documented in the literature and it has mainly been investigated in relation to specific admission diagnoses such as Chronic Obstructive Pulmonary Disease (COPD), heart failure, and orthopedic surgery. For instance, comorbidities have been shown to increase hospital costs for patients admitted with exacerbated COPD [25]. Similarly, having five or more comorbidities is a predictor of rising hospital expenses, as observed in cancer patients [26]. In cases of patients admitted with hip fractures, all

comorbidities—particularly malnutrition and pulmonary thromboembolism—were associated with increased length of stay (LoS) and hospital costs. Likewise, a Charlson Comorbidity Index (CCI) score of 2 or higher was identified as a predictor of higher hospitalization costs for ischemic stroke patients [27]. In contrast to these condition-specific findings, our study demonstrates that the CCI is positively associated with increased hospital costs regardless of the admission diagnosis. Consistent with previous research [28,29], this study confirms the significant impact of comorbidities on hospital costs, reinforcing its role as a robust predictor of overall hospital expenditures. Specifically, each additional point on the Charlson index is associated with a 4.4% increase in costs, while each additional diagnosis at discharge corresponds to a 3.2% rise. Interestingly, Lehnert et al. [28] reviewed multiple studies confirming not only a significant relationship but also a curvilinear, near-exponential relationship between the number of chronic conditions and healthcare expenditures.

Our study identified infectious disorders as the category that is associated with higher hospital costs. This findings aligns with data from the USA Healthcare Cost and Utilization Project [30], which highlighted sepsis as one of the most costly hospital diagnoses. Regarding hospital readmissions, our results confirm that respiratory disorders are a strong predictor of readmission at both 30 days and six months, while circulatory disorders also emerge as a significant risk factor, particularly for readmissions within six months. These findings are consistent with previous research, which has highlighted sepsis, heart failure, diabetes with complications, COPD, and pneumonia as leading causes of hospital readmissions [30]. However, prior studies suggest variability in the influence of the type of diagnosis on readmission risk. For instance, cardiovascular diseases [31], along with heart failure, COPD, and psychosis [24], have been repeatedly identified as the primary conditions contributing to hospital readmissions.

Patients with a higher Barthel Index (BI) at discharge (especially 76+) are less likely to be readmitted within 30 days, but this effect does not persist at six months. The Barthel Index does not significantly impact hospital costs, suggesting that functional status does not drive cost variation. These findings partially reinforce previous research demonstrating that patients with lower BI scores experience increased hospital costs and readmissions [32]. The association between lower functional status and higher hospital costs has also been confirmed using other functional assessment tools, such as the Activities of Daily Living (ADL) index [11]. Additionally, studies have shown that greater dependence, as reflected by a lower BI, correlates with longer length of stay (LoS) during admission [32].

The role of albumin level as a predictor of rehospitalization among the elderly was established by other studies [33,34]. Our study confirms the "protective" role of high albumin levels in reducing short-term readmissions while reinforcing the association between low albumin levels and an increased risk of mortality. Moreover, the findings highlight low albumin levels as an indicator of higher in-hospital costs.

Our study identifies low systolic blood pressure at admission as a predictor of hospital readmission in geriatric patients, regardless of the admission diagnosis. This expands on previous findings by Muzzarelli et al. [35], who reported a similar association but focused specifically on patients with heart failure, where low systolic blood pressure could be part of the severity of the cardiovascular condition. Apart from Muzzarelli et al. [35], our study provides one of the few pieces of evidence supporting the relationship between low systolic blood pressure and rehospitalization, reinforcing the importance of monitoring these readily available vital signs. Beyond its role in predicting readmission, our results also show that low blood pressure at admission is associated with higher hospital costs, though to a lesser extent than other identified variables.

From a clinical perspective, the findings of this study underscore the need for an accurate and comprehensive geriatric assessment in fragile elderly patients to identify those at higher risk of unplanned re-hospitalization. The study sheds light on key patient characteristics, such as the Charlson Comorbidity Index (CCI), disease type, and functional scales, which can help refine healthcare strategies and better understand the financial implications of different patient sub-groups. The need for precise geriatric assessment becomes even more pressing in the context of demographic transition and ageing population in developed countries, which are increasing the pressure on healthcare expenditures, making the demand for cost-effective and high-value interventions particularly crucial in geriatric care. Healthcare systems globally are actively

implementing strategies to reduce hospital readmission rates [10]. The findings of the present study could assist policy-makers and hospital managers in making more cost-effective decisions. For instance, transition care (TC) programs have been demonstrated to be effective in reducing unplanned readmissions for several patient groups, thereby generating cost savings by enhancing outpatient management [36]. However, these programs are highly complex and multidimensional, which means there are various decisions to be made by organizations. These include choosing and number of transitional care (TC) elements to use, the professional expertise needed and the patient (sub-)groups to target [37]. Hospital managers need to make decisions based on understanding which patients are most likely to be readmitted. This means they can decide which transitional care (TC) elements to use, which professionals are needed and which patients to target. This helps to reduce the number of unplanned readmissions and develop cost-effective programs. In addition, in the last years Value-Based Healthcare (VBHC) framework consider the relevant cost of all resources used during a patient's full cycle of care for a specific medical condition, including the treatment of associated complications. VBHC tried to overtake the use of standard charge or top-down cost methodologies in favor of costing techniques that consider specific characteristics of patients. The cost differences among medical conditions and among patients with the same medical condition reveal opportunities to create value [38].

The strength of this research is the integration of diverse data sources, combining administrative records—such as length of stay (LoS), clinical diagnoses, vital signs, and comorbidities—with laboratory findings, including albumin levels, and functional assessments using the Barthel Index (BI). Moreover, unlike many previous studies focusing primarily on short-term outcomes, this research incorporates a longer follow-up period, allowing for a more comprehensive assessment of both early and medium-term readmission risks. This approach provides a more detailed evaluation of patient risk factors and their impact on hospital outcomes. To our knowledge, this study is among the few [11] that combine a large study population with an extended follow-up period. While some studies [11,27] considers larger sample sizes but are limited to a shorter three-month follow-up, others extend their follow-up to 12-months but rely on smaller study samples [32,39].

Some limitations of this study study should be acknowledged. Notably, our study lacks specific data on the primary diagnosis for readmissions. In particular, we were unable to determine whether readmissions were due to the same cause as the index admission or a different one. This prevents a deeper understanding of whether readmissions were related to the original condition or new health issues. Future research should investigate this aspect to better understand the underlying causes of hospital readmissions. Another limitation of this study is the time lag between data collection (2018/2019) and publication. While a timely dissemination of results is ideal, the research process required extensive data preprocessing and integration to ensure accuracy and consistency. Additionally, unexpected challenges in data collection required adjustments to the study's focus, leading us to use the most complete available data for analysis. Despite this delay, the fundamental insights from this study remain highly relevant, as the factors influencing hospital costs and readmissions continue to be critical considerations in current healthcare decision-making.

## Conclusion

This study identifies key predictors of hospitalization costs and unplanned readmissions. Regarding hospitalization costs, our findings align with existing literature, confirming that patients with a longer length of stay (LoS) and a higher number of comorbidities—assessed through both the Charlson index and the number of diagnoses—incur the highest expenses. Additionally, patients recovering from infectious disorders tend to face higher costs compared to those with other major diagnoses.

In terms of readmissions (at 30 days and 6 months), our results show that patients with a prior hospitalization within the previous six months and those with longer LoS have the highest probability of being readmitted at both intervals. Furthermore, patients with higher Barthel Index (BI) scores have a lower risk of readmission at 30 days, though this effect is not observed at 6 months. Conversely, patients with lower albumin levels exhibit a higher probability of readmission.

Future research should explore the potential of biomarkers and functional scales in predicting hospital costs and readmission risk. The findings of this study offer valuable insights for patient prioritization, enabling the development of cost-effective care management strategies aimed at maximizing healthcare benefits.

## Supporting information

**S1 Table. Predictors of hospital costs/health outcomes after 1 year from discharge.**
(PDF)

## Author contributions

**Conceptualization:** Irene Simonetti, Stefano Landi, Chiara Leardini, Anna Giani, Arianna Bortolani, Francesco Fantin.

**Data curation:** Irene Simonetti, Stefano Landi.

**Formal analysis:** Irene Simonetti.

**Methodology:** Irene Simonetti, Stefano Landi.

**Software:** Irene Simonetti.

**Supervision:** Stefano Landi, Chiara Leardini.

**Writing – original draft:** Irene Simonetti, Stefano Landi, Chiara Leardini, Anna Giani, Arianna Bortolani, Francesco Fantin.

**Writing – review & editing:** Irene Simonetti, Stefano Landi, Chiara Leardini, Anna Giani, Arianna Bortolani, Francesco Fantin.

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
