## [Decision Letter · Decision Letter 0]

31 Jan 2025

PONE-D-24-48156Impact of Functional Status and Biomarkers on Hospital Costs and Readmission Rates in Geriatric Patients: An Observational Study with Comprehensive Geriatric AssessmentPLOS ONE

Dear Dr. Simonetti,

Thank you for submitting your manuscript to PLOS ONE. After careful consideration, we feel that it has merit but does not fully meet PLOS ONE’s publication criteria as it currently stands. Therefore, we invite you to submit a revised version of the manuscript that addresses the points raised during the review process.

We look forward to receiving your revised manuscript.

Kind regards,

Pedro Kallas Curiati, M.D., Ph.D.

Academic Editor

PLOS ONE

Journal requirements: When submitting your revision, we need you to address these additional requirements. 1. Please ensure that your manuscript meets PLOS ONE's style requirements, including those for file naming. The PLOS ONE style templates can be found at https://journals.plos.org/plosone/s/file?id=wjVg/PLOSOne_formatting_sample_main_body.pdf and https://journals.plos.org/plosone/s/file?id=ba62/PLOSOne_formatting_sample_title_authors_affiliations.pdf. 2. Please amend either the title on the online submission form (via Edit Submission) or the title in the manuscript so that they are identical. 3. Please amend your list of authors on the manuscript to ensure that each author is linked to an affiliation. Authors’ affiliations should reflect the institution where the work was done (if authors moved subsequently, you can also list the new affiliation stating “current affiliation:….” as necessary). 4. Please provide additional details regarding ethical approval in the body of your manuscript. In the Methods section, please ensure that you have specified the name of the IRB/ethics committee that approved your study. 5. We note that you have indicated that there are restrictions to data sharing for this study. For studies involving human research participant data or other sensitive data, we encourage authors to share de-identified or anonymized data. However, when data cannot be publicly shared for ethical reasons, we allow authors to make their data sets available upon request. For information on unacceptable data access restrictions, please see http://journals.plos.org/plosone/s/data-availability#loc-unacceptable-data-access-restrictions.  Before we proceed with your manuscript, please address the following prompts: a) If there are ethical or legal restrictions on sharing a de-identified data set, please explain them in detail (e.g., data contain potentially identifying or sensitive patient information, data are owned by a third-party organization, etc.) and who has imposed them (e.g., a Research Ethics Committee or Institutional Review Board, etc.). Please also provide contact information for a data access committee, ethics committee, or other institutional body to which data requests may be sent. b) If there are no restrictions, please upload the minimal anonymized data set necessary to replicate your study findings to a stable, public repository and provide us with the relevant URLs, DOIs, or accession numbers. Please see http://www.bmj.com/content/340/bmj.c181.long for guidelines on how to de-identify and prepare clinical data for publication. For a list of recommended repositories, please see https://journals.plos.org/plosone/s/recommended-repositories. You also have the option of uploading the data as Supporting Information files, but we would recommend depositing data directly to a data repository if possible. Please update your Data Availability statement in the submission form accordingly. 6. Please include captions for your Supporting Information files at the end of your manuscript, and update any in-text citations to match accordingly. Please see our Supporting Information guidelines for more information: http://journals.plos.org/plosone/s/supporting-information. 

Reviewers' comments:

Reviewer's Responses to Questions

**Comments to the Author**

1. Is the manuscript technically sound, and do the data support the conclusions?

Reviewer #1: Yes

Reviewer #2: Yes

2. Has the statistical analysis been performed appropriately and rigorously? 

Reviewer #1: Yes

Reviewer #2: Yes

3. Have the authors made all data underlying the findings in their manuscript fully available?

Reviewer #1: Yes

Reviewer #2: Yes

4. Is the manuscript presented in an intelligible fashion and written in standard English?

Reviewer #1: Yes

Reviewer #2: Yes

5. Review Comments to the Author

Reviewer #1: General Evaluation: This study provides a comprehensive and insightful analysis of the factors influencing hospitalization costs and readmissions in elderly patients, focusing strongly on functional status, comorbidities, and biomarkers. Using administrative data combined with clinical variables strengthens the study's findings, and the long follow-up period adds valuable depth to the research.

Strengths:

The study effectively integrates clinical, functional, and laboratory data to predict hospital costs and readmissions, highlighting the significant role of comorbidities, length of stay (LOS), and functional status (Barthel Index).

Identifying low systolic blood pressure as a predictor of readmission is a novel and valuable contribution to the field.

A long-term follow-up period (6 months) and a large cohort enhance the generalizability of the findings.

The discussion thoroughly compares results with existing literature and clearly explains the study's implications.

Areas for Improvement:

Clarity and Structure: To improve readability, the discussion section could benefit from clearer separation between key findings, strengths, and limitations.

Specificity in Findings: More detailed clinical examples or scenarios would enhance understanding of how these findings could be applied in real-world settings.

Comparative Analysis: This study's uniqueness could be emphasized more, particularly its long follow-up period and integration of multiple data sources.

Conciseness: Some sections could be more concise. For instance, simplifying repeated concepts or overly complex sentences could improve clarity without sacrificing detail.

Limitations:

The lack of specific data on the primary diagnosis of readmissions is a notable limitation, as it prevents a deeper understanding of whether readmissions were related to the original condition or new health issues. Future studies could address this gap.

Conclusion: Overall, this study contributes to understanding the determinants of hospital costs and readmissions in elderly patients. With minor revisions for clarity and structure, it would be an excellent addition to the literature.

Reviewer #2: Thanks for the opportunity, I believe that studies that focus on cost reduction associated with detailed knowledge of the population of each country, city, state becomes a facilitator for the development of more accurate public policies and more accurate interventions by health teams. Congratulations on the article.

Update yours reference 36 and adapt the text to the 2024 version:

36. United Nations Department of Economic and Social Affairs, Population Division.

(2019). World population ageing 2019. Highlights. See:

https://www.un.org/en/development/desa/population/publications/pdf/ageing/WorldPo

pulationAgeing2019-Highlights.pdf

The delay in publishing this data, which was collected in 2018/2019, should be explained in the text to justify the reason. I believe this could be a detrimental factor in the context of the relevance of the data to the present day.

Cordialment,

GMA

6. PLOS authors have the option to publish the peer review history of their article (what does this mean? ). If published, this will include your full peer review and any attached files.

**Do you want your identity to be public for this peer review?** For information about this choice, including consent withdrawal, please see our Privacy Policy .

Reviewer #1: No

Reviewer #2: **Yes: ** Guilherme Medeiros de Alvarenga

---

## [Author Response · Author response to Decision Letter 1]

14 Apr 2025

Response Reviewer 1:

We thank Reviewer 1 for the valuable comments. In response to the reviewer’s thoughtful feedback, we have revised and restructured the Discussion section to enhance clarity and coherence by simplifying complex sentences and removing redundant concepts, ensuring that key points remain clear without losing detail.

First, to enhance readability, we have reorganized the Discussion section to clearly separate key findings, strengths, and limitations. The section now begins with a concise discussion of the univariate analysis results, followed by a more detailed explanation of the multivariate analysis findings. This facilitates comparisons with previous studies and reinforces how our work builds upon existing literature.

After presenting our main findings, we have included an extended discussion that incorporates more clinical examples and real-world scenarios. Specifically, we address how demographic shifts and an aging population are placing increasing pressure on healthcare expenditures, making cost-effective interventions in geriatric care more critical than ever. We highlight the role of transition care (TC) programs in reducing unplanned readmissions and generating cost savings by improving outpatient management. Additionally, we discuss how the Value-Based Healthcare (VBHC) framework offers a more patient-centered approach to cost analysis, incorporating disease characteristics and functional assessments to better tailor interventions. These additions provide a broader perspective on how our findings can inform policymakers and hospital managers in optimizing resource allocation and improving healthcare outcomes.

In the final part of the Discussion, we have expanded on the strengths of our study. Specifically, we emphasize the integration of diverse data sources, combining administrative records, laboratory findings, and functional assessments to offer a more comprehensive evaluation of patient risk factors. Additionally, we now clearly highlight the significance of our longer follow-up period, which allows for a more thorough assessment of both early and medium-term readmission risks. Unlike previous studies that either included larger samples with a short follow-up or smaller samples with an extended follow-up, our study balances these aspects, providing a broader perspective on hospital readmissions in geriatric care.

Finally, we conclude the Discussion by outlining the main limitations of the study. These include the lack of specific data on the primary diagnosis for readmissions and the time lag between data collection and publication. This delay stemmed from constraints related to the COVID-19 pandemic as well as the time required for data preprocessing.

Response Reviewer 2:

We thank Reviewer 2 for the valuable comments. First, the opening paragraph of the introduction has been revised to include more recent statistics and references. The reference list has been updated accordingly. Second, we acknowledge the time lag between data collection (2018/2019) and publication as a limitation of this study. Data collection started in 2018, continued through 2019, and was only partially completed in 2020 due to challenges posed by COVID-19. Originally, the data was intended for an observational study on transitional care programs. However, due to the incompleteness of the dataset in 2020, unfortunately, we decided to abandon that study and take advantage of the more complete data from 2018 and 2019 to examine the determinants of hospital costs and readmission. Furthermore, the preprocessing and integration of data into a unified dataset required additional time. This limitation has been acknowledged in the final paragraph of the Discussion Section.

---

## [Editor Report · Decision Letter 1]

25 Apr 2025

Impact of Functional Status and Biomarkers on Hospital Costs and Readmission Rates in Geriatric Patients: An Observational Study with Comprehensive Geriatric Assessment

PONE-D-24-48156R1

Dear Dr. Simonetti,

We’re pleased to inform you that your manuscript has been judged scientifically suitable for publication and will be formally accepted for publication once it meets all outstanding technical requirements.

Kind regards,

Pedro Kallas Curiati, M.D., Ph.D.

Academic Editor

PLOS ONE
---

## [Editor Report · Acceptance letter]

PONE-D-24-48156R1

PLOS ONE

Dear Dr. Simonetti,

I'm pleased to inform you that your manuscript has been deemed suitable for publication in PLOS ONE. Congratulations! Your manuscript is now being handed over to our production team.

Kind regards,

on behalf of

Dr. Pedro Kallas Curiati

Academic Editor

PLOS ONE